# Molecular Clustering Analysis of Blood Biomarkers in World Trade Center Exposed Community Members with Persistent Lower Respiratory Symptoms

**DOI:** 10.3390/ijerph19138102

**Published:** 2022-07-01

**Authors:** Gabriele Grunig, Nedim Durmus, Yian Zhang, Yuting Lu, Sultan Pehlivan, Yuyan Wang, Kathleen Doo, Maria L. Cotrina-Vidal, Roberta Goldring, Kenneth I. Berger, Mengling Liu, Yongzhao Shao, Joan Reibman

**Affiliations:** 1Department of Environmental Medicine, New York University Grossman School of Medicine, New York, NY 10010, USA; 2Division of Pulmonary Medicine, Department of Medicine, New York University Grossman School of Medicine, New York, NY 10016, USA; nedim.durmus@nyulangone.org (N.D.); sultan.pehlivan@nyulangone.org (S.P.); maria.cotrina@nyulangone.org (M.L.C.-V.); roberta.goldring@nyulangone.org (R.G.); kenneth.berger@nyulangone.org (K.I.B.); 3World Trade Center Environmental Health Center, NYC Health + Hospitals, New York, NY 10016, USA; yian.zhang@nyulangone.org (Y.Z.); yuting.lu@nyulangone.org (Y.L.); yuyan.wang@nyulangone.org (Y.W.); mengling.liu@nyulangone.org (M.L.); 4Division of Biostatistics, Department of Population Health, New York University Grossman School of Medicine, New York, NY 10016, USA; 5Pulmonary, Kaiser Permanente East Bay, Oakland, CA 94611, USA; kathleen.x.doo@kp.org

**Keywords:** World Trade Center exposure, plasma biomarkers, lower respiratory symptoms, irritant induced asthma, persistent respiratory symptoms following exposures, September 11 2001 disaster, molecular biomarkers

## Abstract

The destruction of the World Trade Center (WTC) on September 11, 2001 (9/11) released large amounts of toxic dusts and fumes into the air that exposed many community members who lived and/or worked in the local area. Many community members, defined as WTC survivors by the federal government, developed lower respiratory symptoms (LRS). We previously reported the persistence of these symptoms in patients with normal spirometry despite treatment with inhaled corticosteroids and/or long-acting bronchodilators. This report expands upon our study of this group with the goal to identify molecular markers associated with exposure and heterogeneity in WTC survivors with LRS using a selected plasma biomarker approach. Samples from WTC survivors with LRS (*n* = 73, WTCS) and samples from healthy control participants of the NYU Bellevue Asthma Registry (NYUBAR, *n* = 55) were compared. WTCS provided information regarding WTC dust exposure intensity. Hierarchical clustering of the linear biomarker data identified two clusters within WTCS and two clusters within NYUBAR controls. Comparison of the WTCS clusters showed that one cluster had significantly increased levels of circulating matrix metalloproteinases (MMP1, 2, 3, 8, 12, 13), soluble inflammatory receptors (receptor for advanced glycation end-products-RAGE, Interleukin-1 receptor antagonist (IL-1RA), suppression of tumorigenicity (ST)2, triggering receptor expressed on myeloid cells (TREM)1, IL-6Ra, tumor necrosis factor (TNF)RI, TNFRII), and chemokines (IL-8, CC chemokine ligand- CCL17). Furthermore, this WTCS cluster was associated with WTC exposure variables, ash at work, and the participant category workers; but not with the exposure variable WTC dust cloud at 9/11. A comparison of WTC exposure categorial variables identified that chemokines (CCL17, CCL11), circulating receptors (RAGE, TREM1), MMPs (MMP3, MMP12), and vascular markers (Angiogenin, vascular cell adhesion molecule-VCAM1) significantly increased in the more exposed groups. Circulating biomarkers of remodeling and inflammation identified clusters within WTCS and were associated with WTC exposure.

## 1. Introduction

The destruction of the World Trade Center (WTC) towers on September 11, 2001 (9/11) caused acute and/or chronic exposure to dust and fumes with subsequent persistent respiratory symptoms in responders, local residents, and workers [1,2,3,4]. Many community members who were exposed to the toxic dust and fumes in the disaster area of southern Manhattan and Brooklyn developed new onset or aggravated persistent lower respiratory symptoms (LRS) [5,6,7,8]. The WTC Environmental Health Center (WTC EHC) provides treatment and surveillance for these community members, called WTC Survivors (WTCS) under the James Zadroga Health and Compensation Act. Immediate and very intense exposure from the collapse of the WTC towers on September 11, 2001 was followed by the potential for chronic exposure from resuspended dust and fumes while living or working in the area [5]. The alkaline pH (pH 9–10), complex mixture of chemicals, and large size variability contributed to the toxicity of the dust [9]. The exposure, timing of onset of symptoms, type of respiratory symptoms, and bronchial hyperreactivity suggest that these symptoms are consistent with irritant-induced asthma (IIA) [2,3,5,6]. Irritant-induced asthma can be persistent and resistant to full resolution despite treatment [10,11,12,13].

Approximately 15 years after the destruction of the towers, we studied a subgroup of WTC survivors who had persistent LRS (WTCS) with normal spirometry to evaluate responses to optimized treatment for irritant-induced asthma [5]. However, full treatment responses were only seen in a few of the enrolled WTC survivors who showed complex response patterns, including bronchial hyperresponsiveness and co-morbid upper airway symptoms [5]. Plasma samples were obtained during one of the office visits [5], and these samples were analyzed in the current study. The aim of our current investigation was to measure selected, circulating biomarkers to identify molecular markers associated with persistent LRS and WTC exposures. We compared the plasma samples from the WTCS with samples from a non-WTC exposed healthy control group (Controls) derived from the NYU Bellevue Asthma Registry [14,15]. The inclusion of this control group was designed to improve scientific rigor and validity in studying WTC patients. Here, we hypothesized that levels of circulating biomarkers would (a) distinguish WTCS and Controls, (b) identify WTCS heterogeneity and (c) find associations with WTC exposure. We included biomarkers of inflammation and repair with an approach that has been used to study outcomes in other lung and cardiovascular conditions [16,17,18,19]. Studies in WTC Responders have demonstrated the importance of blood-derived inflammatory and metabolic molecular markers for predicting disease onset and progression [20,21,22,23,24,25,26,27,28,29,30]. We used biomarkers of remodeling, including matrix metalloproteinases (MMPs), because of their role in irritant-induced asthma [10]. Accordingly, we assayed MMPs and the tissue inhibitor of metalloproteinases (TIMP1), circulating receptors of inflammation and metabolism including the receptor for advanced glycation end-products (RAGE), triggering receptor expressed on myeloid cells (TREM1), circulating Interleukins (IL), tumor necrosis factor (TNFα) and their soluble receptors, chemokines, vascular mediators (angiogenin, angiopoietin 2, vascular cell adhesion molecule- VCAM1), and lung derived mediators (surfactant protein D-SPD).

## 2. Materials and Methods

### 2.1. Ethical Considerations

The study was approved by the NYU Grossman School of Medicine institutional review board. All study participants signed informed consent. The approval numbers are S 13-00448 for the Uncontrolled Lower Respiratory Symptoms in the WTC Survivor Program which is also listed at ClinicalTrials.gov, identifier NCT02024204; and H-9698 for the NYUBAR registry (Disease Modifying Genes in Severe Asthma), which is also listed at ClinicalTrials.gov, identifier NCT00212537.

### 2.2. Patients

WTCS were defined as patients from the WTC EHC who had participated in the original study [5]. The clinical details of the participants who donated plasma samples that were analyzed here have been previously reported [5]. WTCS were included if they had lower respiratory symptoms (LRS) and their asthma symptoms were uncontrolled, defined as an asthma control test (ACT) score at the initial visit equaled or was less than 20 signifying little control, and their lung function as measured by spirometry was within normal limits [5]. Some WTCS participants with a history of LRS and controlled asthma symptoms, with an asthma control score >20 were also subsequently included in the original study and were used for this analysis as well [5]. The following were exclusion criteria [5]: a > 5 pack-year history of tobacco use, cardiac disease, active cancer, other chronic lung diseases (sarcoidosis, interstitial lung disease, bronchiectasis), asthma predating 9/11, abnormal spirometry at study visit 1, and abnormalities on the lung radiographs (Chest X-ray, CT). A total of 73 WTCS participants were enrolled [5].

Spirometry was performed before and after administration of a bronchodilator to determine pre- and post-bronchodilator Forced Vital Capacity (FVC) and pre- and post-bronchodilator Forced Expiratory Volume in 1 s (FEV1) [5]. The spirometry data were converted to % predicted based on normative values [31].

For control group comparison, plasma samples from the NYU Bellevue Asthma Registry (NYUBAR) [14,15] were randomly chosen (*n* = 55). This registry was created to study patients with asthma as well as controls and details have been published previously [14,15]. The controls from the NYUBAR were selected to match the WTCS based on age (on September 11, 2001), gender, BMI, and spirometry (Table 1).

### 2.3. Samples

The original clinical trial was not a biomarker study [5]. Therefore, one blood sample was obtained. Blood samples were collected on one occasion, and due to technical issues, most participants had samples obtained at study visit 1 (*n* = 43), however, some had samples obtained at visit 2 (*n* = 2), or at the end of the study (visit 4, *n* = 28). Plasma samples were immediately prepared, aliquoted, and stored frozen at −80 °C until analysis. We were able to test the samples from all of the 73 WTCS participants using multiplex assays for 34 biomarkers (Appendix A). However, in less than 10% of the samples the assay failed and this caused data variability with *n* = 67–72 for TNFα, IL-33, IL-1β, IL-6, surfactant protein D (SPD), CC chemokine ligand (CCL)11, IL-8 (leukocyte), Glycoprotein (GP130), IL-6Ra, TNFRI, TNFRII, MMP3, MMP8, MMP7, MMP12, MMP13; *n* = 64 for MMP2 and TIMP1; *n* = 73 for all other analytes. The hierarchical clustering algorithm only included samples with a full biomarker dataset (no missing data) with an *n* = 58 for the WTCS samples.

For the NYUBAR controls, we obtained data for most analytes from *n* = 55 samples. *n* = 54 for MMP13; *n* = 52 for TNFα, IL-1β, IL-6, IL-33; *n* = 47 for MMP2, and TIMP1. The hierarchical clustering algorithm only included samples with a full biomarker dataset (no missing data) with an *n* = 41 for the NYUBAR controls.

### 2.4. Multiplex Assays

The multiplex assays were performed per manufacturers’ protocols with premixed magnetic microsphere kits from R&D Systems (Minneapolis, MN, USA; catalogue numbers LXSAHM regular, or FCSTM14 high sensitivity) or Millipore (Merck, Darmstadt, Germany; catalogue number HSTCMAG-28SK high sensitivity) with a FlexMAP3D instrument (Luminex, Austin, TX, USA) and xPONENT software (Luminex, Austin, TX, USA).

Most runs were performed with 1 or 2 plates each (40 or 88 samples). The sample dilution was at least 1:2, and up to 1:2000 as indicated in Appendix A. For each run, a multiplexed standard curve was assayed in duplicate for each of the analytes. A negative, multiplexed duplicate control was also included. The upper and lower limits of the standard curve ranges are shown in Appendix A. Samples were assayed in duplicate in the dilutions indicated (Appendix A).

The sample concentrations were calculated based on the standard curves. For each standard curve concentration (pg/mL, *x*-axis, 7 serially diluted standards) the mean of the mean fluorescence intensity was calculated (*y*-axis). Interpolation from log-transformed concentrations (pg/mL) and log transformed mean fluorescence intensity values was performed using asymmetric sigmoidal, 5PL, least squares fit interpolation as instructed by each kit and programming of the xPONENT software that controls the multiplex Flexmap3D instrument.

Run-to-run comparability of the data was established by carefully randomizing samples on each plate (e.g., WTCS vs. controls). Furthermore, technically, the run-to-run comparability was ensured by testing random duplicate samples on random plates. This strategy was used for the determination of CRP, Angiogenin, ST2, MMP1, CHI3L1/YKL-40, Angiopoietin-2, IL-1RA, CCL20/MIP-3α, RAGE, Periostin, VCAM1, CCL17/TARC, TREM1, MMP2, TIMP1, TNFα, IL-33, IL-1ß, IL-6, IL-13, IL-4, IL-8. For the IL-13, IL-4, IL-8 kit, additional internal standards were tested in duplicate at 2 concentrations in each run. For the TNF RII/TNFRSF1B, TNF RI/TNFRSF1A, SP-D, MMP3, MMP7, MMP8, MMP12, MMP13, IL-8/CXCL8, IL-6R alpha, GP130, CCL11/Eotaxin kit, we did not have duplicate samples or additional standards to ensure optimal run-to-run compatibility; however, we observed that for each analyte, the mean fluorescence intensity across the samples was comparable for each of the 7 runs used to test all of the samples. Therefore, for each analyte, we selected 3 standard curves that most optimally covered the full range of mean fluorescence data generated by all the samples to calculate a single composite standard curve. The standard concentration (X) was plotted against the 3 values for the mean of the mean fluorescence intensity (Y) of the 3 selected standard curves with their 7 serially diluted data points. This composite standard curve was used to perform interpolation of log-transformed concentrations (pg/mL) and log transformed mean fluorescence intensity values using asymmetric sigmoidal, 5PL, least squares fit to calculate the concentration of each analyte in each of the 128 samples analyzed [32,33,34].

For all assay runs and analytes, the standard curves showed high goodness of fit with the adjusted R-squared values being at least 0.95, and for most assay runs 0.99 or more, and for some runs 1.0.

The finalized analyte concentration (pg/mL) in each sample considered the dilution factor. For samples in which values were outside of the range of the standard curve, either below or above the detection range, we generated numeric values, either at least 2-fold below the limit of detection, or 2-fold above the detection range. For an analyte that had measured values below the detection limit in some of the samples, for example with a detection limit of 1 pg/mL all measurements below the limit (not detected) were given the numerical value of 0.05 pg/mL. For an analyte that had measured values above the detection range in some of the samples, for example with upper detection value of 12 µg/mL, the samples above this limit were given the numerical value of 25 µg/mL.

### 2.5. Statistical Analysis

Median and interquartile range were used to summarize continuous variables. Count and percentage were used to summarize categorical variables. Hierarchical clustering was performed with an algorithm that had no missing values from all of the 34 biomarker analytes, with samples from WTCS (*n* = 58) and NYUBAR control (Control, *n* = 41). Hierarchical clustering on biomarker values was performed using the method Ward from linear, standardized biomarker data using the JMP Pro16.2 software (SAS Institute Inc. 2020–2021. JMP® 16, Cary, NC, USA: SAS Institute Inc.). The hierarchical clustering was performed with samples (listed in rows) on biomarker values (listed in columns). The biomarker values were standardized by subtracting the column means and divided by the column standard deviations. The resulting WTCS clusters (WTCS cluster 1, WTCS cluster 2) were compared for biomarker analyte levels with logistic regression to adjust for age, body mass index (BMI), gender, and race/ethnicity on log transformed biomarker concentrations. WTC exposures were considered as categorical variables: WTC dust cloud exposure on the day of 9/11 (no/yes), workplace with ash (no/yes), WTCS participant category (worker/resident). The variable workers included local workers and clean-up workers. The variable resident included residents and others. We also had information on other WTC exposure categories (for example ash at home (no/yes)), but there was not enough data variability or sample size to perform group comparisons. Data for some categorical variables contained some missing data causing variability in the actual sample size used in specific statistical analyses. Multivariable logistic regression was also used to assess significance of association between WTC exposure status and the two WTCS clusters with adjustment for age, BMI, gender, and race/ethnicity. Bivariate comparisons between WTCS clusters 1 and 2 for selected categorical variables including race/ethnicity, gender, exposures categories (worker/resident) and workplace ash status (yes/no) were also performed with the Fisher’s exact test. Group comparisons (SPSS version 28 software, IBM Corp. Released 2021. IBM SPSS Statistics for Macintosh, Version 28.0. Armonk, NY, USA: IBM Corp) were also performed with two-tailed, unpaired non-parametric Mann–Whitney test. *p* values of <0.05 were considered significant. The calculations were performed with the R software (R core team, 2013, Vienna, Austria). Graphs were produced with JMP 16.2 Pro (SAS Institute Inc. 2020–2021. JMP^®^ 16, Cary, NC, USA: SAS Institute Inc.) or Graphpad Prism (GraphPad Prism for MacOS, Version 9.3.1 (350), December 7, 2021, GraphPad Software, San Diego, CA, USA, www.graphpad.com).

## 3. Results

### 3.1. Demographic Characteristic of Study Populations

The demographic characteristics of the WTCS and the NYUBAR control participants are shown in Table 1. The clinical characteristics of the WTCS have been described in detail previously [5]. WTCS were older than NYUBAR controls and there were slight differences in race, but no difference was noted for pre and post bronchodilator spirometry.

### 3.2. Biomarker Analysis and Hierarchical Clustering

The levels of circulating biomarkers in WTCS plasma samples were within the same overall range as the circulating biomarker levels in the controls as shown by the summary analysis in Appendix A. For example, an analyte (CCL11) with median detection level of 115 pg/mL in WTCS samples, had a similar median level (92 pg/mL) in NYUBAR control samples. Additionally, for samples from both the WTCS group and from the NYUBAR control group, we noticed variability in the measurements. Therefore, hierarchical clustering was performed to further understand the distributions of the WTCS samples and NYUBAR control samples (Figure 1). Linear, standardized biomarker values were used in the clustering protocol. The data showed that there were 2 clusters that contained nearly exclusively WTCS samples, and 2 clusters that contained nearly exclusively NYUBAR control samples. The clusters were comprised of 50 of 58 WTCS samples, and 30 of 41 NYUBAR control samples. Appendix A show the full dendrogram of the participants and analytes, respectively. Appendix A shows the demographic characteristics of the WTCS clusters.

### 3.3. Comparison of WTC Clusters 1 and 2 for Biomarker Levels

To characterize the biomarker levels in WTCS clusters 1 and 2 samples, the clusters were compared for the concentration of the circulating analytes by non-parametric statistics and false discovery rate on log-transformed data, and additionally, by logistic regression to adjust for age, BMI, gender, and race/ethnicity on log transformed biomarker levels (Figure 2). The comparisons showed significant differences among the clusters within MMP biomarkers (MMP1, 2, 3, 8, 12, 13), soluble receptors (RAGE, IL-1RA, ST2, TREM1, IL-6Ra, TNFRI, TNFRII) and also with chemokines (IL-8-leukocyte, CCL17), Figure 2.

Differences in race/ethnic groups were identified between the two clusters (Appendix A). To further confirm that biomarkers, and not age or race/ethnicity, played dominating roles in the clustering, we showed that a few of these significant biomarkers were not associated with age and race/ethnicity and could form logistic models with a very high area under the curve (AUC) to distinguish/classify the two WTCS clusters (Appendix A). Specifically, the two WTCS clusters were compared with multivariable logistic regression and ROC (receiver operating characteristic)/AUC (area under the ROC curve) analysis to determine potential influences of age and race/ethnicity on the circulating biomarker levels. Among the 15 biomarkers that were significantly different between WTCS clusters 1 and 2 (with adjustment for age, BMI, gender and ethnicity, Figure 2), 8 biomarkers (MMP1, 8, 12, 13, IL-1RA, TNFRI, TREM1, IL-6Ra) were the least correlated with age (−0.1 < correlation coefficient < 0.1). Using the 8 biomarkers to fit a logistic regression to classify cluster 1 and cluster 2, the AUC was 0.985 (Appendix A). As we had identified 8 biomarkers that were not associated with age, we then analyzed these 8 biomarkers for association with race/ethnicity. In total, 3 (MMP8, MMP12, TNFRI) of the 8 biomarkers showed no significant association with race/ethnicity by linear regression between biomarker levels and race/ethnicity. The *p*-values were >0.05 for each category of race/ethnicity and each biomarker. When MMP8, MMP12 and TNFRI were used to fit a logistic regression to classify WTCS cluster 1 and cluster 2, the AUC was 0.907 (Appendix A).

### 3.4. Comparison of WTC Clusters 1 and 2 for WTC Exposures

To test the hypothesis that the WTCS clusters would be distinguished by WTC dust exposure, we analyzed categorical variables of exposure. WTCS cluster 2 reported significantly more frequently on workplace with ash exposure than WTCS cluster 1 (Figure 3A). Furthermore, WTCS cluster 2 had more workers than residents when compared to WTCS cluster 1 (Figure 3B, *p* < 0.05). Logistic regression was used to adjust for the co-variates age, BMI, gender, and ethnicity/race (Figure 3A,B). In contrast, the immediate, high-level exposure by the WTC dust cloud at September 11, 2001 (9/11) variable was not significantly different between cluster 1 and 2 (*p* = 0.773, Figure 3C). There was also no difference between clusters 1 and 2 with respect to dust or ash at home exposure, however only few participants (5 of 40 who provided answers) described heavy dust or ash at home.

Because we had differences in time of blood draw and clinical symptoms within the WTCS, we also compared the WTCS clusters based on the designation of controlled or uncontrolled LRS described in the clinical study [5] and for the study visit during which the blood samples were obtained. We found no significant differences between the WTCS clusters based on clinical status or office visit of the blood draw (Appendix A).

### 3.5. Comparison of WTC Exposure Categories for Biomarker Levels

Significant differences between WTCS cluster 1 and cluster 2 in circulating biomarkers (Figure 2) and WTC exposure categories (Figure 3) suggested the possibility that circulating biomarker levels would differ between groups defined by categorical WTC exposure variables. To address this question, groups were determined by WTC exposure variables—(a) dust cloud at September 11, 2001 (9/11, Figure 4A, no vs. yes); (b) participants (Figure 4B, resident vs. worker); (c) workplace with ash (Figure 4C, no vs. yes)—were compared for biomarker levels using the Mann–Whitney test. Biomarkers had significantly different levels between groups, which were determined by WTC exposure variables (Figure 4). They consisted of a chemokine (CCL17/TARC), soluble receptors (RAGE, TREM1), and MMP12 (Figure 4A,B). These markers were also increased in WTCS cluster 2 when compared with WTCS cluster 1 (Figure 2). Furthermore, we found increased levels of vascular markers (angiogenin, VCAM1, Figure 4A,B) in groups defined by variables of increased WTC exposure. Additionally, levels of a chemokine CCL11/eotaxin and MMP3 were significantly increased in the group of WTC dust cloud at 9/11 exposure—yes, relative to the group WTC dust cloud at 9/11 exposure—no (Figure 4C). The groups defined by WTC exposure variables were not different with respect to age, BMI, gender, or race/ethnicity (Appendix A).

We also compared clinical status of LRS (controlled, *n* = 19 vs. uncontrolled, *n* = 53–54) designated as previously described [5] and biomarker levels. We identified four biomarkers that were significantly different (lower) in the WTCS whose LRS was controlled; specifically, the soluble receptors GP130 (*p* = 0.037), IL-1RA (*p* = 0.013), TREM1 (*p* ≤ 0.001) and the vascular marker VCAM1 (*p* = 0.016). The comparisons were performed with the Mann Whitney test. There was no difference in the groups of WTCS defined by LRS status with respect to age and BMI (*p* > 0.05 Mann–Whitney test), or gender and ethnic group (*p* > 0.05, Fisher’s exact test).

## 4. Discussion

We studied WTCS and a non-WTC exposed control group (NYUBAR controls) with similar spirometry values. We used a WTCS group with persistent LRS and normal spirometry who had participated in a previous study [5], where we showed that these symptoms persisted despite the use of high dose inhaled corticosteroids and long acting beta2 agonists. Furthermore, we studied WTC unexposed controls because it is generally agreed in the WTC research community that incorporating non-WTC exposed cohorts can improve scientific rigor and validity of the study. Hierarchical clustering of circulating biomarkers identified two clusters of WTC survivors separated by two distinct clusters of the non-WTC controls. The separate clusters of WTCS and non-WTC groups in unsupervised biomarker-based clustering indicates potentially that a specific group of biomarkers at specific level ranges might be associated with WTC exposure status that is unique to the WTCS group. This idea was further strengthened by showing that the clusters were determined by specific biomarkers, and not age or race/ethnicity using ROC analysis. Additionally, we identified biomarkers that were associated with WTC exposure. Our group and others have previous established association of WTC exposures with LRS, cancer, mental health disorders, neuropathic symptoms, and cognitive declines in WTC survivors [35,36,37,38,39,40,41,42,43,44,45,46].

The identification of well-separated clusters among the WTCS participants or the NYUBAR control participants was not surprising because previously published work had shown circulating markers of allergy in both groups. Among the WTCS participants nearly ½ had circulating, allergen specific IgE [5], and approximately ½ of the NYUBAR control participants have been reported to have atopy [15]. While the allergen specific IgE previously detected in the WTCS by the clinical study [5] and the signs of atopy in the NYUBAR control participants [15] constitute signs of T helper 2 (type 2) immune responses, the targeted biomarker approach that we used was not designed to specifically identify immune response types. Instead, our choice of MMP’s, soluble receptors, chemokines, cytokines, and other mediators was designed to detect processes of tissue remodeling, inflammation, and aging that are important in persistent respiratory and cardiovascular diseases [14,16,18,19,23,47] and in WTC responders [22,25,26,27,28,29].

For the interpretation of our data, it is important to know if the values of the biomarkers measured by our assays correspond to published levels of analytes, particularly in control participants. In part, concentrations of circulating biomarkers depend on the medium analyzed, plasma vs. serum [48,49,50]. Searching for published data on plasma biomarker analytes, we were able to find examples for a substantial subset the biomarkers tested in our study. They include MMP1, 2, 3, 8, 12, 13, TIMP1 [17,18,19], IL-6 and its receptors IL-6Ra, GP130 [16], receptors in the IL-1 family: IL-1RA [51], ST2 [17], TNFa [18], TNFRI [52], TNFRII [52], the chitinase YKL40 [14,17], the pattern recognition receptors TREM1 [47] and RAGE [53], the vascular markers VCAM1 [18], angiogenin [54], angiopoietin 2 [54], the tissue remodeling, non-structural matrix protein Periostin [55], the general inflammation marker CRP [8,18], and the chemokine CCL17 [56]. The cytokines IL-4, IL-13, IL-6, IL-33, or the chemokine IL-8 were detected in the expected low pg/mL range.

WTCS cluster 2 was characterized by circulating inflammatory and remodeling markers including MMP1, 2, 3, 8, 12, 13, soluble receptors RAGE, IL-1RA, IL-6Ra, TNFRI, TNFRII, and chemokines IL-8, and CCL17. Many of these biomarkers, in particular MMP’s and RAGE, have previously been reported to be very important to understand the responses in the WTC responders, signifying either a prediction of protection or worsening of lung function [18,23,24,25,26,57]. Our data cannot predict lung function measured by spirometry because participants were enrolled based on normal spirometry measurements ([5], Table 1). As detailed in the manuscript [5] that describes the clinical data of the WTCS whose plasma was analyzed in the current study, the asthma control score did not improve in the majority of the study participants [5]. Similarly, we found that the WTCS clusters were not associated with the LRS clinical control status, or with the office visit during which the plasma sample was obtained. This may also explain why the study of WTC responder firefighters identified additional biomarkers that were not different among the two clusters identified in our studies here. Those biomarkers either predicting protection or worsening of lung function include a chitinase YKL40 [58,59], the cytokines IL-6 and TNFα [60,61].

The WTC responder analysis also identified very important metabolic, vascular and additional inflammatory [20,62,63,64,65,66,67] components that determine persistent respiratory symptoms due to the exposure. We focused on a targeted biomarker approach for the current study, which has well-established precedents in cardiovascular or lung health [28,30]. The two clusters of WTCS were compared with statistical adjustment for body mass index, and also age, gender and race/ethnicity. Furthermore CRP, a marker which is reported to be associated with overweight [68], was not different between the WTCS clusters before adjustment for BMI (*p* = 0.9). These results do not exclude the possibility that molecular metabolic shifts occurred in WTCS participants from either cluster 1 or 2; this needs to be investigated in future studies.

While we cannot directly compare biomarker levels between our study that was performed with plasma and the published data from WTC responder firefighters, which was performed with serum [18,19,24,25,26], our data provide an important extension of the already gained knowledge in the WTC responder program. Our study extends knowledge regarding gender in the responses to the WTC collapse. In contrast to the studies with WTC responder firefighters [18,19,24,25,26] that had mostly male participants, the WTCS in our study were mostly females, including in both clusters. Our clusters identified by biomarker levels were significantly different based on WTC dust exposure, an insight based on data from initial visiting questionnaire (IVQ) and follow up studies. Additionally, our study identified biomarkers (CCL17, RAGE, TREM1, MMP12) that differed between groups of WTCS clusters and WTC exposure categorical variables. Furthermore, we identified additional vascular markers, a chemokine, and a MMP marker to be increased in samples from WTCS exposed to increased WTC dust (assessed by categorical variables). Together, these data suggest that an intensive and chronic WTC dust exposure induced an injury that persisted for 15 years, likely by a continuing process of incomplete repair. In addition, our data likely reflect aging processes. A soluble receptor marker (TREM1) and a vascular marker (VCAM1) that were associated with WTC dust exposure were significantly higher in samples of WTCS who had uncontrolled LRS status when compared to WTCS who had controlled LRS status. This finding suggests the possibility that the clinical control status of LRS might possibly be linked to the intensity of the WTC dust exposure. The studies in the WTC responder cohort were conducted with samples that were drawn approximately 3/4 year following the WTC collapse [18,19,24,25,26]. The differences in the timing of the biomarker data relative to WTC dust exposure emphasize that the WTC responder studies have been predictive [18,19,24,25,26], while our study is reflective, looking back on the exposure with WTC dust.

Ethnic determinants of genetic diversity [69] and of determinants of circulating protein biomarkers of the type that we investigated here have been previously highlighted [54,70]. Of particular interest for our own data is the finding that ethnic polymorphisms at the APOE (apolipoprotein E) locus are major determinants of metabolism and are associated with ethnically different plasma protein levels of MMP3 [70]. MMP3 is part of the major MMP gene cluster on chromosome 11q [71,72] that also includes MMP1, 8, MMP12, and MMP13, all of which we found to be present at significantly higher levels in cluster 2 WTCS participants. We also found significant differences among the two clusters of WTCS in MMP2 (adjusted *p* = 0.015). The MMP2 gene is not located in the major MMP cluster on chromosome 11, but on chromosome 16 [72]. The finding that MMP2 was also significantly different among clusters together with the adjustment of all *p* values for race/ethnicity, BMI, age, gender when comparing the clusters gives us confidence that our findings do not simply reflect variations due to ethnicity.

It is possible that the persistence of toxins acquired during the WTC exposure drives the persistence of processes reflected by the increased biomarkers seen in our study. Dr. Shao’s and Dr. Trasande’s groups have shown persistent increases, more than 12 years after 9/11, in serum dioxins and furans in WTC exposed children who experienced dust at home [73]. Initially, within the first months after the WTC tower collapse, increased fire related chemicals (that include dioxins and furans) were detected in WTC responder firefighters [74], National Guard responder workers [75], and pregnant women (WTC survivors) [76]. The persistent markers of WTC exposure of children in the affected community is of particular concern [77,78,79,80,81,82,83,84]. The WTC dust exposures may have induced epigenetic reprogramming [36] that caused the alterations in circulating biomarkers that we detected, and this needs further investigation.

Alternatively, it is possible that the WTC exposure led to de novo development of aberrant inflammatory responses, such as an increase in allergic and asthmatic responses described in children exposed to WTC dust [85], in WTC rescue and clean-up workers [86], and suggested in children exposed to outdoor air pollution [87]. Some of the bio-markers that we found to be increased in the WTC categorical variables of more intense exposures, include markers that can be associated with T helper 2 (type 2) responses, for example CCL17/TARC, CCL11/eotaxin [88], MMP12 [89], or VCAM1 [90]. Other markers, like RAGE [89], TREM1 [91], or angiogenin (and also VCAM1 [92]), can be associated with Th17 (type 17) responses. These findings could support the idea that the WTC dust exposures of the WTC survivors triggered aberrant immune responses similar to the conditions described before in WTC dust exposed children and responders [85,86] as asthmatic responses frequently have type 2 and also type 17 components [93]. However, these same markers can also signify remodeling and cancer development where MMP’s and chemokines cooperate [94], while RAGE may signify a metabolic shift by which inflammation is regulated [95]. Future studies should focus on distinguishing between these possibilities.

The asthma control score did not improve in the majority of the study participants [5] and study participants had different levels of symptom control [5], a finding that may have contributed to the differences in the clusters. However, we found that the WTCS clusters were not associated with the LRS clinical control status, or with the office visit during which the plasma sample was obtained. We cannot pinpoint the organ or cells that produce the bio-markers at increased levels in WTCS cluster 2 participants, or in the groups of increased WTC exposure. It is possible that the biomarkers are produced in the lungs, but other organs, for example the skin or the upper airways, may also contribute. This remains to be determined in future studies. Careful risk benefit analyses prompted us to study plasma samples that are obtained by a low-risk procedure. Therefore, we decided against sampling the lungs directly for example by bronchoalveolar lavage. Our study is also limited by the volunteer participants of the clinical study [5] whose samples we analyzed here, and by the selected biomarkers that were measured. Establishment of a bio-bank combined with proteomic studies would give us more detailed and unbiased data. In future investigation, we would like to further assess the impacts of age and race/ethnicity, as well as BMI and gender on WTCS biomarkers and clinical data, with a much larger sample size and prospective longitudinal blood samples. Studies that would directly compare WTC responders and WTC survivors now 20 years after the WTC destruction for biomarkers of repair and aging would also be highly informative. Future studies would also need to address potential interactions between biomarker levels, WTC dust exposure, inflammation, repair/aging, co-morbidities, treatment, and nutrition [96]. Our study represents a step towards the overall goal to better understand the molecular changes associated with the respiratory condition in the WTC survivors with persistent LRS that could inform future optimized and personalized treatment.

## 5. Conclusions

In conclusion, our study expands upon previous studies of blood biomarkers in WTC exposed populations using a sample of WTCS with ongoing LRS years after exposure. We provide evidence for an association between intensity of WTC exposures and blood-based biomarkers of inflammation, repair, and aging in WTC survivors. More specifically, our study identified heterogeneity (clusters) among the WTC survivors with LRS based on circulating biomarkers 15 years following the disaster. These biomarkers could reflect WTC exposure, persistent inflammation, ongoing repair, and aging processes. A better understanding of the molecular changes associated with the respiratory condition in the WTC survivors could be used to devise future personalized treatment.

## Figures and Tables

**Figure 1 ijerph-19-08102-f001:**
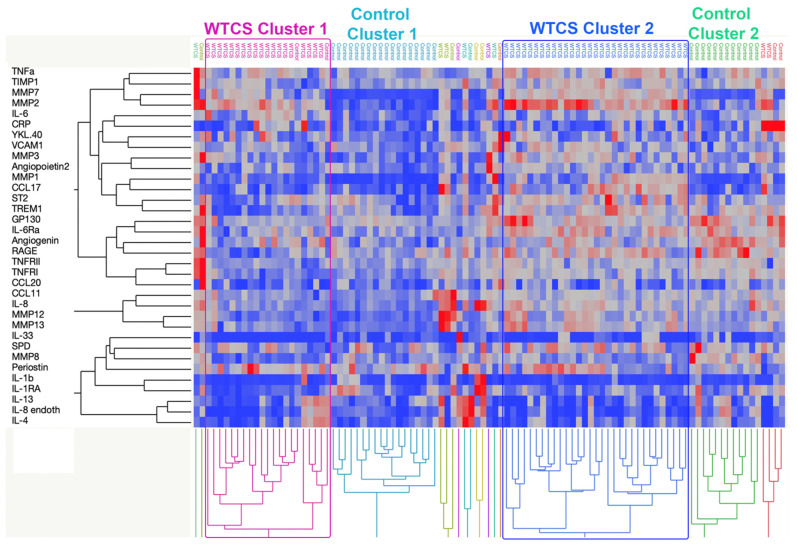
Hierarchical clustering of WTC survivor (WTCS, *n* = 58) and NYUBAR control (Control, *n* = 41) samples by biomarker analytes. Clustering was performed using the method Ward from linear, standardized biomarker data. WTCS clusters 1 and 2, as well as control clusters 1 and 2, are indicated. The colors signify lowest (dark blue) to highest value (bright red). The dendrograms are abbreviated here and shown in full in Appendix A.

**Figure 2 ijerph-19-08102-f002:**
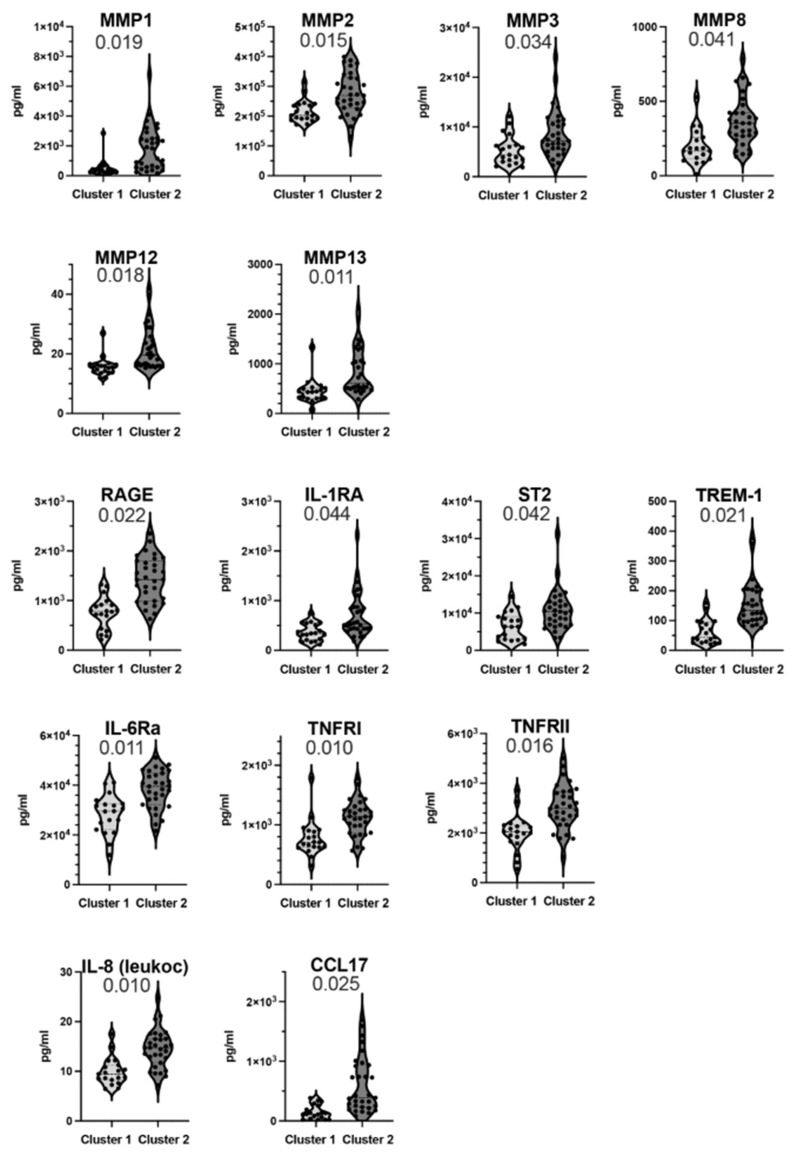
Violin plots show the comparison of WTC survivor clusters (cluster 1, cluster 2) for levels of biomarker analytes. Significant differences among clusters were identified in MMP markers, soluble receptors, and chemokines. These biomarker analytes are shown, the abbreviations are CCL: CC chemokine ligand, IL: Interleukin, MMP: matrix metalloproteinase, R: receptor, RAGE: receptor for advanced glycation end-products, ST2: suppression of tumorigenicity 2, TNF: tumor necrosis factor, TREM1: triggering receptor expressed on myeloid cells 1. IL-8 levels were determined with an assay (RnD Systems) that used leukocyte IL-8 to generate the detection reagents. Cluster 1 (*n* = 19) and cluster 2 (*n* = 31) were compared by logistic regression adjusting for age, BMI (body mass index), gender, and race/ethnicity on log transformed biomarker levels. *p* values are indicated on top of each plot.

**Figure 3 ijerph-19-08102-f003:**
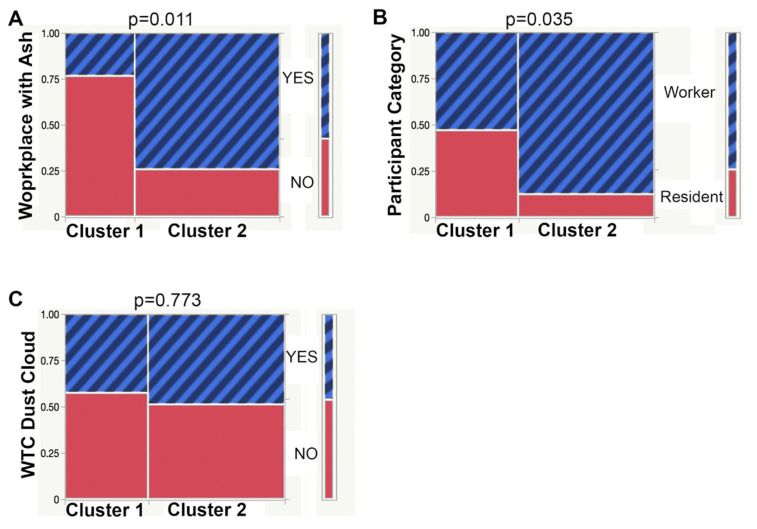
Mosaic plots show the comparison of WTC survivor clusters (cluster 1, cluster 2) for the WTC exposure categorical variables, (**A**) Workplace with ash (no/yes), (**B**) Participant category (resident/worker), (**C**) WTC dust cloud at 9/11 (no/yes). Cluster 1: *n* = 13–19, cluster 2: *n* = 27–31 were compared by logistic regression adjusting for age, BMI, gender, and race/ethnicity. *p* values are indicated on top of each plot.

**Figure 4 ijerph-19-08102-f004:**
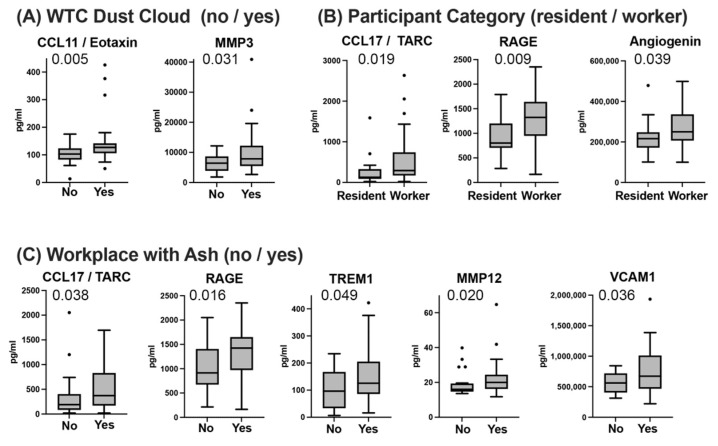
Comparison of WTC exposure variables with respect to biomarker analytes. (**A**) WTC Dust Cloud at September 11, 2001 (9/11), no: *n* = 36, yes: *n* = 37; (**B**) participant category, residents: *n* = 16, workers: *n* = 57; (**C**) workplace with Ash, no: *n* = 20–21, yes: *n* = 30–37. The box plots were drawn according to Tukey, biomarkers that were significantly different between groups are shown. *p*-values were calculated with the independent, two-tailed Mann–Whitney test. Age, BMI, gender, or race/ethnicity were not different between groups defined by exposure categories (Appendix A).

**Table 1 ijerph-19-08102-t001:** Characteristics of all study participants.

	Level	NYUBAR Control ^1^	WTCS	*p*	Test	Missing (%)
Age, mean (SD)		40.2 (11.4)	55.6 (10.2)	<0.001		0
Gender, *n* (%)	F	35 (63.6)	51 (69.9)	0.581		0
	M	20 (36.4)	22 (30.1)			
BMI, mean (SD)		28.9 (6.8)	30.3 (5.4)	0.219		0
Ethnicity, *n* (%)	Black	9 (16.4)	20 (27.4)	0.046	exact	0
	Latino	36 (65.5)	33 (45.2)			
	White	10 (18.2)	15 (20.5)			
	White Other	0 (0)	5 (6.8)			
Education, *n* (%)	Grade school (up to 6th grade)	2 (3.6)	3 (4.2)	0.903		0.8
	High school (12th grade)	18 (32.7)	26 (36.1)			
	More than high school	35 (63.6)	43 (59.7)			
PrePre BD FVC % pred ^2^ (mean (SD))		93.1 (11.3)	95.4 (15.5)	0.352		0.8
Post BD FVC % pred ^2^ (mean (SD))		92.1 (11.7)	97.0 (14.9)	0.047		2.3
Pre BD FEV_1_ % pred ^3^ (mean (SD))		93.1 (11.3)	93.1 (13.7)	0.981		0.8
Post BD FEV_1_ % pred ^3^ (mean (SD))		94.1 (12.0)	96.3 (13.9)	0.363		2.3
IgE (IU/mL), mean (SD)		147.3 (212.3)	106.3 (201.3)	0.268		0
WTC Exposure	WTC dust cloud	0/55	73/73			
Data ^4^, *n*/N	Workplace with ash	0/55	58/73			
	Participant Category	0/55	73/73			

^1^ NYUBAR Controls [14,15]; ^2^ Pre- or post-bronchodilator (BD) forced vital capacity (FVC), % predicted (pred) [31]; ^3^ pre- or post-bronchodilator (BD) forced expiratory volume in 1 s (FEV_1_), % predicted; ^4^ available data points.

## Data Availability

All data are provided in the Tables, Figures, Appendix A.

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
