# Peer review of "Molecular Clustering Analysis of Blood Biomarkers in World Trade Center Exposed Community Members with Persistent Lower Respiratory Symptoms"

_ijerph, 2022, doi:10.3390/ijerph19138102_

Round 1
Reviewer 1 Report
The manuscript “Molecular Clustering Analysis of Blood Biomarkers in World 2 Trade Center Exposed Community Members with Persistent 3 Lower Respiratory Symptoms” is a very interesting subject, focusing on the consequences of a major disaster with human exposure to dust and fumes from the WTC collapse. The written English should be revised and there are some issues that need to be revised and improved, as follows:
Comments:
Line 110: The participants provided only 1 sample? And why is there this disparity in sample collection, with some participants providing in the earlier visits, others in the last visit and none in the third? Explore a little more about the 4 visits, the populations recruitment and the sample collection, please.
Does the difference in age and race between the WTCS and controls interfere with the results?
Author Response
The authors thank the reviewer for the constructive and thoughtful comments.
English: the text has been carefully revised
-Line 110: The participants provided only 1 sample? And why is there this disparity in sample collection, with some participants providing in the earlier visits, others in the last visit and none in the third? Explore a little more about the 4 visits, the populations recruitment and the sample collection, please.
The participants provided only one sample at the visits indicated. The clinical aspects of this study have been previously published in detail (Dr. Reibman and colleagues, 2020), and this study has now been carefully discussed throughout the current manuscript. The biomarker analysis described here was made possible as an ancillary study to the clinical treatment study (Reibman and colleagues, 2020), and therefore, only one blood sample was taken. As outlined in the clinical manuscript, only a few participants achieved controlled lower respiratory symptoms (LRS) classification, and most participants remained at the uncontrolled LRS status. Accordingly, we report here that the timing of the sample acquisition (visit 1, 2 at the beginning of the study, or 4 at the end of the study and treatment), or the LRS control status were not significantly associated with the WTC survivor clusters. This is better explained in the text. The data regarding the comparison of the WTC survivor clusters with the timing of blood draw or the clinical status of LRS control are shown in the new Supplemental Figure S5.
- Does the difference in age and race between the WTCS and controls interfere with the results?
All comparisons between controls and WTCS, and particularly between clusters of the WTCS have been statistically corrected for age, BMI, and race/ ethnicity. This is shown graphically for the WTCS clusters in the new Supplemental Figure S4. The comparisons between controls and WTCS are represented in Supplemental Fig. S1 and show that the levels measured in the samples of the WTCS were within the ranges measured in the samples from the control group. Using hierarchical clustering analysis of the biomarkers (Figure 2), we found two separate clusters of WTCS and two separate clusters of controls. We then compared the two WTCS clusters (adjusting for age, BMI, gender and ethnicity/race), but did not compare the control clusters with the WTCS clusters. One explanation for the clustering of the controls is that approximately 1/2 of the participants were atopic, and the other 1/2 non-atopic. This explanation is presented in the discussion section.
The overall hypothesis of the manuscript has been further clarified to indicate that the current study was focused on the comparison of the groups of WTCS and controls, the analysis of the WTCS group clusters, and the association of biomarkers with categorical variables of WTC exposure. "Here we test the hypotheses that levels of circulating biomarkers would be similar in WTCS and Controls, that the WTCS were a homogeneous group with respect to plasma biomarkers, and that the circulating biomarkers and WTC exposure were not associated". All three hypotheses were rejected by our data.
Reviewer 2 Report
Major Comments
1- The authors analyzed blood samples to assess the cytokine profile of patients with respiratory symptoms. Why evaluate the systemic circulation rather than analyzing the bronchoalveolar lavages, knowing that it is the lungs that are exposed in the first line.
For COVID-19, for example, it has recently been demonstrated that the portrait of inflammatory molecules measured in the blood does not mirror that in the lungs in COVID-19.
2-A lack of clinical information prevents the reader to draw solid conclusions... maybe the observed results are influenced by comorbidities and/or medication. Please provide additional demographic information and clinicobiological data.
3-Why the following molecules: Fraktalkine, G-CSF, GM-CSF, IL-12p40, IL12p70, IL15, IL17A, CXCL9... were not added to the panel ?
Author Response
The authors thank the reviewer for the careful and constructive comments.
1- The authors analyzed blood samples to assess the cytokine profile of patients with respiratory symptoms. Why evaluate the systemic circulation rather than analyzing the bronchoalveolar lavages, knowing that it is the lungs that are exposed in the first line.
We performed a careful risk benefit analysis and it was decided that plasma sample analysis was rather non-invasive, and involved less risk than direct sampling of the lungs and therefore, plasma samples were obtained for our study. It is of note, that the same safety considerations have also applied to the WTC responders, where biomarker analysis of blood samples have been published. We acknowledge that BAL samples would have given us more specific insight into the lungs. However, the safer and less invasive blood sampling was chosen. This has been clarified in the discussion section.
2-A lack of clinical information prevents the reader to draw solid conclusions... maybe the observed results are influenced by comorbidities and/or medication. Please provide additional demographic information and clinicobiological data.
The current biomarker study is an ancillary study to a clinical study that has been described in detail by Dr. Reibman and colleagues (2020). This has been much better explained throughout the current manuscript. As explained in the manuscript describing the clinical study (Reibman and colleagues 2020), only a few WTC survivors were classified as having achieved controlled LRS, while most of the WTC survivors in this study remained at uncontrolled LRS, despite of treatment.
A comparison of the WTCS clusters identified in the present study based on biomarker levels with respect to controlled vs uncontrolled LRS status did not reveal significant differences and this has been presented in Supplemental Figure S5. Furthermore, a comparison of the WTCS clusters with respect to the visit number at which the plasma sample was obtained (visit 1 was at the beginning of the study, visit 4 was at the completion of the study and treatment) is also presented in Supplemental Figure S5. Again, there was no significant difference between the WTCS clusters. Furthermore, we identified 4 biomarkers (3 soluble receptors, and 1 vascular marker) which were found to be significantly different (significantly decreased) between groups of WTC survivors whose LRS was controlled vs. uncontrolled. These markers are identified in the results section and discussed in the discussion section.
3-Why the following molecules: Fraktalkine, G-CSF, GM-CSF, IL-12p40, IL12p70, IL15, IL17A, CXCL9... were not added to the panel ?
We would have loved to have been able to add these markers to the study and specifically, we would have liked to optimally perform a proteomic analysis. This would have enabled us to obtain a much more detailed view into molecular processes. However, we did not have the funds necessary and therefore performed an analysis on a limited number of analytes. This has been stressed throughout the manuscript.
Reviewer 3 Report
the paper have interesting research framework and can be regarded as good case study report the introduction is satisfactory and methodology gives enough information for the reader about experiments. authors analyse the influence of WTC leftover on residents and visitors health. i have just few concern that authors should better explain first why the number of women was much higher than men participants, and if in study dominant was latino people can we make conclusion on whole research group.
Author Response
The authors thank the reviewer for the careful and insightful comments.
I have just few concern that authors should better explain first why the number of women was much higher than men participants, and if in study dominant was latino people can we make conclusion on whole research group.
This is a limitation of the current study, we were able to analyze blood samples from volunteers who participated in the clinical treatment study that has been described in detail by Dr. Reibman and colleagues (2020). The biomarker analysis presented here was an ancillary study and therefore the WTCS study participants of the clinical study (Reibman and colleagues 2020) donated one sample that we were then able to analyze.
This is why our study is representative of this previously published clinical study because we were able to obtain biomarker results, depending on the analyte from all or nearly all samples. However, we would love to establish a large plasma biobank that is representative of the WTC survivor community as a whole, and proteomics studies would be possible. With that biobank we would be able to perform a much more representative study. Furthermore, we would be able to compare our study to other proteomics studies regarding lung health or responses to environmental exposures sponsored for example by the NHLBI. We have now indicated this idea in the discussion section. The ultimate goal of this research is to understand the molecular mechanisms of the WTC dust exposure induced disease better, to devise personalized treatment options. The current study is a step towards this goal. We have emphasized this in the discussion section.
Furthermore, we show the comparison of the clusters with respect to gender, ethnic groups, age and BMI in the new Supplemental Figure S4.
Round 2
Reviewer 2 Report
Replies to my comments have been addressed correctly